# Compensatory Modulation of Seed Storage Protein Synthesis and Alteration of Starch Accumulation by Selective Editing of 13 kDa Prolamin Genes by CRISPR-Cas9 in Rice

**DOI:** 10.3390/ijms25126579

**Published:** 2024-06-14

**Authors:** Hue Anh Pham, Kyoungwon Cho, Anh Duc Tran, Deepanwita Chandra, Jinpyo So, Hanh Thi Thuy Nguyen, Hyunkyu Sang, Jong-Yeol Lee, Oksoo Han

**Affiliations:** 1Kumho Life Science Laboratory, Department of Integrative Food, Bioscience and Biotechnology, College of Agriculture and Life Sciences, Chonnam National University, Gwangju 61166, Republic of Korea; phamhueanh199@gmail.com (H.A.P.); kw.cho253@gmail.com (K.C.); anhduchytq1997@gmail.com (A.D.T.); dpllchandra@gmail.com (D.C.); thwlsvy123@gmail.com (J.S.); hksang@jnu.ac.kr (H.S.); 2Faculty of Biotechnology, Vietnam National University of Agriculture, Hanoi 12406, Vietnam; ntthanh.sh@vnua.edu.vn; 3Department of Agricultural Biotechnology, National Institute of Agricultural Science, RDA, Jeonju 54874, Republic of Korea

**Keywords:** prolamin, 13 kDa prolamin, glutelin, seed storage protein, CRISPR-Cas9, protein body, starch, rice seed, ER stress, differentially expressed genes

## Abstract

Rice prolamins are categorized into three groups by molecular size (10, 13, or 16 kDa), while the 13 kDa prolamins are assigned to four subgroups (Pro13a-I, Pro13a-II, Pro13b-I, and Pro13b-II) based on cysteine residue content. Since lowering prolamin content in rice is essential to minimize indigestion and allergy risks, we generated four knockout lines using CRISPR-Cas9, which selectively reduced the expression of a specific subgroup of the 13 kDa prolamins. These four mutant rice lines also showed the compensatory expression of glutelins and non-targeted prolamins and were accompanied by low grain weight, altered starch content, and atypically-shaped starch granules and protein bodies. Transcriptome analysis identified 746 differentially expressed genes associated with 13 kDa prolamins during development. Correlation analysis revealed negative associations between genes in Pro13a-I and those in Pro13a-II and Pro13b-I/II subgroups. Furthermore, alterations in the transcription levels of 9 ER stress and 17 transcription factor genes were also observed in mutant rice lines with suppressed expression of 13 kDa prolamin. Our results provide profound insight into the functional role of 13 kDa rice prolamins in the regulatory mechanisms underlying rice seed development, suggesting their promising potential application to improve nutritional and immunological value.

## 1. Introduction

Seed storage proteins (SSPs), which are the nitrogen source for developing plant seedlings, are classified into four groups based on their solubility: water-soluble albumins, saline-soluble globulins, dilute acid-/alkali-soluble glutelins, and alcohol-soluble prolamins [1,2,3]. In rice, glutelins account for 60–80% of all SSPs and are encoded by 15 genes classified into four groups based on amino acid sequence similarity: GluA, GluB, GluC, and GluD [4,5]. Prolamins account for 20–30% of SSPs, are encoded by 34 genes [6], and are classified into three groups based on their molecular weight: 10, 13, or 16 kDa. The major group of prolamins are 13 kDa in size, and this group is further divided into four subgroups based on cysteine content: Pro13a-I, Pro13a-II, Pro13b-I, and Pro13b-II [7,8].

SSPs are synthesized at the rough endoplasmic reticulum (ER), translocated to the ER lumen, and then transferred to other intracellular compartments of the plant endomembrane system [9,10]. Prolamins accumulate in the ER-derived protein bodies (PB) I (PB-I), whereas globulins and glutelins accumulate in PB-II structures before subsequently moving to protein storage vacuoles (PSVs) [10,11]. The PB-I particle has a layered structure, with the 10 kDa prolamin in the core (innermost) layer, the Pro13b-I subgroup in the layer surrounding the core, 16 kDa and Pro13a-I/II subgroups in the middle layer, and Pro13b-II subgroup prolamins in the outer-most layer [8].

Multiple transcription factor (TF) genes regulate and coordinate biosynthetic pathways for starch and SSPs in the rice endosperm [12]. TFs specifically involved in rice prolamin gene expression include rice prolamin box-binding factor (*RPBF*), rice seed basic leucine-zipper 1 (*RISBZ1*), and NAM, ATAF, and CUC TFs (*OsNAC20* and *OsNAC26*) [13,14,15]. *RPBF*, a member of the DNA binding with one finger (DOF) protein family TFs, activates glutelin and prolamin gene expression by binding to the “P box” motif in target gene promoters [13]. *RISBZ1* interacts with *RPBF* and activates expression of SSP synthesis-related genes GluA1, GluA2, GluA3, GluB1, GluD1, 10 kDa prolamin, 13 kDa prolamin, and 16 kDa prolamin [14]. *OsNAC20/26* are also important regulators of starch and other SSPs, particularly Pul, GluB4, α-globulin, and 16 kDa prolamin [15].

Rice prolamins are deficient in lysine and methionine, two of the nine amino acids considered to be essential in the human diet [2,16,17]. In addition, prolamins are considered indigestible and nutritionally inferior to glutelins [18,19]. Several studies demonstrated that rice SSPs modulate glutathione metabolism, mitigate oxidative damage to lipids and proteins, and provoke antioxidative responses against hypercholesterolemia, although glutelins appeared to be superior to prolamins as antioxidants [19,20,21]. Moreover, prolamins found in grains such as wheat, barley, and rye are known to cause allergic reactions such as celiac disease and IgE-mediated food allergies [22,23,24]. It is interesting that amino acid sequence analysis has revealed a close proximity between prolamins and protein allergens (*RA16/17*) in rice seeds [25,26]. Furthermore, previous studies have shown that the production of recombinant proteins is more efficient in mutant rice lacking 13 kDa prolamin (13 kDa Pro-less rice) than in wild-type (WT) control rice [27,28]. Consequently, there is significant interest in rice lines engineered with low prolamin content, based on the idea that such mutant rice lines will facilitate the production and distribution of recombinant and/or therapeutic proteins and that the grain of such rice lines will be nutritionally superior for humans. Furthermore, engineered mutant rice lines that lack specific prolamin genes are excellent experimental tools for understanding how prolamins and other SSPs influence rice grain development, composition, and properties. Such knowledge will improve our understanding of rice biology and will inform future efforts to improve rice as an agriculturally valuable plant species.

Many published studies demonstrate that the suppression of 13 kDa prolamin expression using RNA-silencing [4,27] and RNA-interference [2,29] was accompanied by compensatory changes in the expression of other SSPs as well as ER stress and the expression of stress-related proteins. ER stress was associated with increased expression of protein folding-related genes, such as binding protein (BiP), protein disulfide isomerase (PDI), and calnexin (CNX) [30,31,32,33]. BiP, a key ER chaperone that belongs to the HSP70 family, interacts with and promotes the proper folding of prolamins [34,35]. PDI facilitates disulfide bond formation and rearrangement in glutelins and prolamins to promote maturation and the proper folding of SSPs during seed maturation. CNX binds to unfolded glycoproteins, preventing misfolded proteins from being transported from the ER to the Golgi apparatus [25,32,36]. As mentioned above, transgenic rice lacking 13 kDa prolamins has increased levels of glutelin, globulin, BiP, and PDI [4]. Similarly, in another study, RNAi-treated transgenic rice with low levels of 13 kDa prolamin showed compensatory accumulation of 10 kDa prolamin and glutelins, up-regulation of chaperone proteins such as BiP and PDI, and abnormal PB-I bodies [2]. In addition, GPGb-RNAi knock-down (targeting glutelins, prolamins, and globulins) generated a rice line with low levels of glutelin A (GluA), cysteine-rich 13 kDa prolamin, and α-globulin, with higher accumulation of mRNA and protein for glutelin B (GluB) and chaperones (BiP1 and PDIL1-1) [29]. A few studies targeted multiple glutelin genes by CRISPR-Cas9 and produced mutant rice lines with significantly reduced glutelin content accompanied by an alteration of starch structure [37,38]. We previously suggested that rice glutelin genes may interact with genes that regulate the synthesis of starch and other seed storage proteins and modulate their expressions via post-transcriptional and translational mechanisms [38]. However, the editing of the prolamin gene by CRISPR-Cas9 has never been reported. In this study, CRISPR-Cas9 was employed to target the Pro13a-I and Pro13b-I/II subgroups of 13 kDa prolamin genes and generated four mutant lines in rice, and their seeds were characterized in order to elucidate the role of specific 13 kDa prolamin genes in seed development and how deletion of 13 kDa prolamin affects seed characteristics, starch composition, and the organization of protein storage compartments during endosperm maturation. Our results indicated compensatory changes in the rice proteome and transcriptome and revealed alterations affecting ER stress response, transcription factor, and starch synthesis in mutant rice with a selectively edited 13 kDa prolamin gene.

## 2. Results

### 2.1. Design of sgRNA for Targeting 13 kDa Prolamin Genes Using CRISPR-Cas9 Technology

There are four genes encoding 10 kDa prolamins, 28 genes encoding 13 kDa prolamins, and two genes encoding 16 kDa prolamins in the rice genome. The 13 kDa prolamin genes are divided into four subgroups (Pro13a-I, Pro13a-II, Pro13b-I, and Pro13b-II) based on cysteine content and sequence similarity (Figure 1A). The nucleotide sequence similarity between *Pro13a.1* and *Pro13a.2* of the Pro13a-I subgroup is 92.03%, while these two genes share 75.33 to 78.17% similarity with the four genes in the Pro13a-II subgroup (*Pro13a.3~6*), which are highly similar to each other (97.93 to 99.68%). Lastly, the Pro13b-II subgroup includes 18 prolamin genes, which demonstrate 87.59 to 100% sequence similarity to each other. The high similarity between 13 kDa prolamin genes in each subgroup indicates that multiple prolamin genes can be edited using one sgRNA. We designed two sgRNAs, as shown in Appendix A and Appendix A. sgRNA-pro13a targets two genes in the Pro13a-I subgroup, and sgRNA-pro13b targets 17 genes in the Pro13b-I/II subgroups. An in vitro assay was used to demonstrate that the sgRNAs efficiently and specifically induce cleavage of their target genes (Appendix A). Subsequently, sgRNA-pro13a and sgRNA-pro13b were cloned into a binary vector for the CRISPR-Cas9 system (Figure 1B and Appendix A). Each pCAMBIA-Cas9-sgRNA binary vector was then introduced into the Korean rice cv. Ilmi (*Oryza sativa* subsp. *Japonica*) via *Agrobacterium*-mediated transformation to generate *13 kDa prolam*in-knockout rice.

### 2.2. Generating and Characterizing 13 kDa Prolamin-Knockout Rice

PCR analysis was performed using *SpCas9* and *HygR* primers specific to the pCAMBIA-Cas9 vector to determine the presence of the binary vector transformed in the T_0_ lines. Results showed that 8 and 10 T_0_ transgenic rice plants were transformed, with editing genes inserted in the target loci of the Pro13a-I and Pro13b-I/II subgroups, respectively (Appendix A). Next-generation sequencing was carried out to identify mutations in target genes in two T_0_ transgenic plants (*1a* and *2a*) in the Pro13a-I subgroup and two T_0_ transgenic plants (*4b* and *8b*) in the Pro13b-I/II subgroups (Appendix A and Appendix A). The results showed that genes in the Pro13a-I subgroup were mutated in the *1a* and *2a* T_0_ plants at frequencies of 51.3 and 28.5%, respectively, and genes in the Pro13b-I/II subgroups were mutated in the *4b* and *8b* T_0_ plants at frequencies of 4.8 and 12.6%, respectively.

To obtain mutant plants without inserted foreign genes in the T_1_ generation, PCR was used to screen for the absence of *HygR* and *SpCas9* genes. This identified 4 (*1a-6*, -*8*, -*10*, and -*11*) and 5 (*2a-1*, -2, -6, -7, and -9) plants in the *1a* and *2a* lines, respectively. Similarly, 2 (*4b-3*, and -*9*) and 3 (*8b-3*, -*6*, and -*11*) plants were identified in the *4b* and *8b* lines, respectively (Appendix A). Furthermore, DNA sequence data showed that *Pro13a.1* and *Pro13a.2* genes are mutated in *1a-6*, -*8*, and -*11* plants, while the *Pro13a.1* gene is mutated in *2a-1*, -2, -7, and -9 plants. In the lines targeting genes in the Pro13b-I/II subgroups, the *Pro13b.1* and *Pro13b.13* genes were mutated in the *4b-9* plant, and the *Pro13b.3* gene was mutated in the *8b-3* and *-11* plants (Appendix A). Targeted deep-sequencing was conducted, and the data was used to select four representative homozygous T_2_ generation plant lines for further study, where the lines were designated *1a-8-1*, *2a-2-1*, *4b-9-7*, and *8b-3-9*. More specifically, the *Pro13a.1* and *Pro13a.2* genes in the *1a-8-1* plant line are mutated at the frequencies of 99.5 and 97.3%, respectively; the *Pro13a.1* gene in the *2a-2-1* plant line is mutated at 99.1%; the *Pro13b.1* and *Pro13b.13* genes are mutated in the *4b-9-7* plant line at the frequencies of 98.1 and 99.8%, respectively; and the *Pro13b.3* gene is mutated in the *8b-3-9* plant at 97.8% (Figure 2A and Appendix A).

### 2.3. Target Gene Expression at Transcriptional and Translational Levels

The expression of prolamin-encoding transcripts was compared in the immature seeds of *13 kDa prolamin*-knockout T_2_ and WT control plants 2 weeks after flowering (2 WAF). The results (Figure 2B) show that transcripts of mutant *Pro13a.1* and *Pro13a.2* genes were remarkably decreased in *1a-8-1* plants; transcripts of mutant *Pro13a.1* gene were strongly reduced in *2a-2-1* plants; transcripts of mutant *Pro13b.1* and *Pro13b.13* genes were decreased in *4b-9-7* plants; and transcripts of mutant *Pro13b.3* were decreased in *8b-3-9* plants. These results are consistent with expectations based on the targeted mutations introduced into these four mutant plant lines. At the translational level, SSP content was analyzed in the mature seeds of T_2_ plants by SDS-PAGE and western blot. SDS-PAGE analysis did not show dramatic differences between SSP content/expression in the four mutant plant lines and WT control plants. However, western blot analysis showed that prolamins in Pro13a-I/II subgroups were strongly and weakly decreased in mature seeds of *1a-8-1* and *2a-2-1* plants, respectively, due to the number of mutated genes. Similarly, expression of Pro13b-I/II subgroups were strongly and weakly decreased in mature seeds of *4b-9-7* and *8b-3-9* plants, respectively (Figure 2C).

### 2.4. Morphological Features of 13 kDa Prolamin-Knockout Plants

Phenotypic features of de-husked seeds from *13 kDa prolamin*-knockout and WT plants were compared, including appearance, length, width, thickness, ratio of length to width, and total weight of 100 grains (Figure 3). In the *pro13a.1/13a.2*-knockout (*1a-8-1*) and *pro13a.1*-knockout (*2a-2-1*) plants, grain length, width, and thickness were notably smaller than WT. However, in the *pro13b.1*/*13b.13*-knockout plant (*4b-9-7*) and *pro13b.3*-knockout plants (*8b-3-9*), the only substantial difference from WT was in grain width (Figure 3C). The morphological differences between the seeds of *13 kDa prolamin*-knockout mutant lines and WT were relatively small and were not large enough to explain the fact that the weight of 100 mutant seeds was 19.8, 34.2, 27.1, or 15.5% lower than WT, respectively, for seeds from *1a-8-1*, *2a-2-1*, *4b-9-7*, *and 8b-3-9* mutant plants (Figure 3B).

To explore possible explanations for the difference between mutant and WT average weight per 100 seeds, scanning electron microscopy (SEM) was used to examine seed endosperm structure, starch structure, and starch composition. When seeds were viewed in cross-section, opaque “chalky” areas were observed (Figure 4A). Furthermore, starch granules in WT seeds exhibited a closely packed, dense pattern of uniform polyhedrons, while starch granules in mutant seeds displayed an irregular arrangement and a loosely packed structure (Figure 4A). The starch content was approximately 25.7, 38.0, 36.5, and 20.1% lower in mutant seeds from *1a-8-1*, *2a-2-1*, *4b-9-7*, and *8b-3-9* plants, respectively, than in WT control seeds (Figure 4B). This suggests that the lower average weight of 100 seeds from the *13 kDa prolamin*-knockout lines is likely caused by low starch content.

Furthermore, analysis of the amylose and amylopectin content of starch granules revealed a lower ratio of amylose to amylopectin in *1a-8-1*, *2a-2-1*, and *8b-3-9* plant seeds than in WT, and a higher ratio of amylose to amylopectin in *4b-9-7* plant seeds than in WT (Figure 4C). The starch granules formed angular blocky structures in *1a-8-1*, *2a-2-1*, and *8b-3-9* plant seeds (low amylose content), while starch granules formed spherical structures in *4b-9-7* plant seeds (high amylose content) (Figure 4A). qRT-PCR was performed in WT and mutant plant seeds to determine the level of expression of ten genes involved in starch metabolism, including ADP-glucose small and large subunit pyrophosphorylases (*AGPS2* and *AGPL2*), starch synthases (*SSI, SSIIa*, and *SSIIIa*), granule-bound starch synthase I (*GBSSI*), branching enzymes (*BEI, BEIIa*, and *BEIIb*), and debranching isoamylase (*ISA1*) (Figure 4D). The results showed that *SSI* and *BEIIa* are significantly down-regulated in all four mutant lines. However, *GBSSI*, implicated in amylose synthesis, was down-regulated in *1a-8-1*, *2a-2-1*, and *8b-3-9* mutant plants but up-regulated in *4b-9-7* plants. These differences in gene expression probably cause the low average starch mass per grain, altered starch composition, and altered morphology of seeds in the mutant plant lines.

### 2.5. Protein Body Formation and SSP Composition in 13 kDa Prolamin-Knockout Lines

Transmission electron microscopy (TEM) was used to investigate PB-I formation and subcellular structures in the immature seeds collected (two WAF) from WT and *13 kDa prolamin*-knockout T_2_ plants. In WT, PB-I structures form by the sequential accumulation of concentric layers composed of Pro10, Pro13b-I, Pro13a-I/II, Pro16, and 13b-II prolamins outward from the core, thus exhibiting a distinct pattern of electron-dense spherical layers (Figure 5A). In contrast, in *pro13a.1*/*13a.2*-knockout (*1a-8-1*) and *pro13a.1*-knockout (*2a-2-1*) plants, PB-I particles were smaller, and the layered structure was entirely lacking. On the other hand, in *pro13b.1/13b.13*-knockout (*4b-9-7*) and *pro13b.3*-knockout (*8b-3-9*) plants, PB-I particles stained more darkly in the outer layers and core than WT PB-I particles. This indicates that the 13 kDa prolamins are critical for forming layered PB-I particles in the seed endosperm of WT rice.

Furthermore, the total content of SSP per grain was higher in *1a-8-1* plant seeds than in WT, while the total SSP content per grain was similar in WT and the other mutant plant lines (Figure 5B). In contrast, prolamins (all sizes) represented 15.8, 15.3, 14.8, and 17.5% of all SSPs in *1a-8-1*, *2a-2-1*, *4b-9-7*, and *8b-3-9* mutant plants, respectively, while prolamins represented 20.0% of WT SSPs (Figure 5C). Conversely, the percent content of glutelins increased significantly from 60.2% in WT to 73.9, 70.0, 69.8, and 65.8% in *1a-8-1*, *2a-2-1*, *4b-9-7*, and *8b-3-9* mutant plant lines, respectively. These results indicate that suppression of expression of certain SSPs was accompanied by compensatory increases in the expression of other SSPs; more specifically, down-regulation of genes in the Pro13a-I subgroup was accompanied by up-regulation of genes in the Pro13a-II, Pro13b-I/II, GluB, and GluD SSP subgroups. Similarly, decreased transcription of Pro13b-I subgroup prolamins was accompanied by increased transcription of Pro13a-II, Pro10, and GluB SSPs. Nevertheless, down-regulation of genes in Pro13b-I/II subgroups was not accompanied by compensatory changes in expression of SSP genes in other groups (Figure 5D).

### 2.6. Identification of DEGs in Immature Seeds of 13 kDa Prolamin-Knockout Plants

To confirm and extend the above results, RNA-seq analysis was performed using RNA from WT and four mutant plants. The results showed that 391 (268 up, 123 down), 372 (194 up, 178 down), 217 (117 up, 100 down), and 394 (175 up, 219 down) genes were differentially expressed in *1a-8-1*, *2a-2-1*, *4b-9-7*, and *8b-3-9* mutant plants, respectively, relative to WT. Gene ontology (GO) functional analysis of the differentially expressed genes (DEGs) was performed using the MapCave tool from MapMan software (http://mapman.gabipd.org/web/guest/mapcave (accessed on 10 February 2024)). A total of 540 up-regulated genes were categorized into functional groups, including RNA processing (25.5%), protein synthesis ribosome RNA (11%), development (9.2%), stress (7.1%), RNA transcription and RNA regulation of transcription (6.1%), miscellaneous (6.1%), transport (3.7%), and other minor categories. In addition, 403 down-regulated genes were categorized into functional groups, including RNA processing (30.1%), protein synthesis transfer RNA (10.9%), protein activity regulation-related (7.9%), RNA transcription and RNA regulation of transcription (7.4%), miscellaneous (6.6%), photosynthesis (5.2%), transport (4.4%), development (3.5%), and other minor categories (Figure 6 and Appendix A). Ten up-regulated and nine down-regulated DEGs were commonly found in all four mutant lines (Figure 6). Among up-regulated DEGs, three genes (ENSRNA049465003, ENSRNA049469812, and ENSRNA049472132) were involved in “RNA processing” and one gene (Os03g0266900) was associated with “stress”. Meanwhile, out of down-regulated DEGs, three genes (ENSRNA049445795, EPlORYSAT000373615, and EPlORYSAT000373835) were related to “protein synthesis transfer RNA”, two genes (ENSRNA049467117 and ENSRNA049471778) were involved in “RNA processing”, and one gene (Os12g0242100) was associated with the “cell wall”. Functions of the rest of the genes are unknown.

### 2.7. Correlation Network Involving DEGs and 13 kDa Prolamins

To investigate patterns of transcriptional change among prolamin genes in the immature seeds of WT, *1a-8-1*, *2a-2-1*, *4b-9-7,* and *8b-3-9* plants, Pearson correlation analysis was performed among 927 DEGs and prolamin genes. This analysis identified 746 transcripts with a correlation coefficient ≥|0.7| (Appendix A). We then constructed a correlation network based on GO enrichment categories such as RNA processing (BIN27.1), RNA transcription and RNA regulation of transcription (BIN27.2 and 27.3), protein synthesis ribosome (BIN29.2.1 and 29.2.6), protein synthesis transfer RNA (BIN29.2.7), protein synthesis elongation (BIN29.2.4), protein folding (BIN29.6), protein degradation (BIN29.5), protein posttranslational modification (BIN29.4), protein targeting (BIN29.3), stress (BIN20), and transport (BIN34) (Figure 7). Furthermore, the results revealed that *Pro13a.1* and *Pro13a.2* genes in the Pro13a-I subgroup are negatively correlated with genes in the Pro13a-II and Pro13b-I/II subgroups. Genes in the network formed six clusters with distinct patterns of correlation with prolamin genes, as follows: Cluster 1 shows positive correlation with genes in the Pro13a-II and Pro13b-I/II subgroups; cluster 2 demonstrates an inverse association with cluster 1, displaying a negative correlation with genes in the Pro13a-II and Pro13b-I/II subgroups; cluster 3 exhibits a positive correlation with the *Pro13a.1* gene but a negative correlation with genes in the Pro13a-II and Pro13b-I/II subgroups, whereas its opposite is cluster 4; similarly, cluster 5 shows a positive correlation with *Pro13a.2* gene and a negative correlation with genes in the Pro13a-II and Pro13b-I/II subgroups, with cluster 6 being its opposite (details in Appendix A). The network reveals which genes are correlated with each prolamin gene. Thus, the results suggest that transcriptional regulatory mechanisms may exist and may explain how and why altered expression of some prolamins is accompanied by compensatory changes in expression of other prolamins and other network genes.

### 2.8. Analysis of Endoplasmic Reticulum Pathway Genes

Of the genes involved in the “Protein processing in the endoplasmic reticulum” pathway, nine genes play roles in ER-associated degradation processing (ERAD). These include *HSP70* (Os03g0276500, Os05g0460000, and Os01g0840100) in cluster 1 and *OsDjC35* (Os03g0776900), *HSP90* (Os04g0107900), *HSP26* (Os03g0245800), *HSP20* (Os11g0244200), *HSP17.7* (Os03g0267200), and *HSP22a* (Os04g0445100) in cluster 3. Three *HSP70* genes in cluster 1, which are positively correlated with prolamin genes in the Pro13a-II and Pro13b-I/II subgroups, were up-regulated in *2a-2-1* and *8b-3-9* mutants. Six genes in cluster 3, which are positively correlated with the *Pro13a.1* gene but negatively correlated with the *Pro13a.3* and prolamin genes in the Pro13b-I/II subgroups, exhibited increased expression in *4b-9-7* mutants (Figure 8A). qRT-PCR analysis of the expression of *HSP20, HSP70*, and *HSP90* revealed a strong correlation between RNA-seq and qRT-PCR estimates of the expression of these genes (see Pearson’s correlation plot in Appendix A). Furthermore, the expression of four genes, *BiP-1, PDIL-1-1, PDIL-2-3*, and *CNX*, which are associated with misfolded protein repair, was also examined (Figure 8B). The results showed that transcription of *BiP-1* and *CNX* genes was consistently 1.5 to 2-fold higher in the immature seeds of the *13 kDa prolamin*-knockout lines than in WT plants. However, the expression of *PDIL1-1* was higher in *2a-2-1* and *8b-3-9* plants than in WT, while the expression of *PDIL2-3* was higher in *1a-8-1* and *4b-9-7* plants than in WT, showing no consistent trend.

### 2.9. Role of Transcription Factors in Regulating the 13 kDa Prolamin Correlation Network

We identified 17 TF genes in clusters 1, 2, 3, and 4 of the correlation network shown in Figure 7 and Appendix A. In particular, the expression of seven TF genes in cluster 1, encoding heat stress TF (*OsHsfB2b*, Os08g0546800), MADS-box TF (*OsMADS26*, Os08g0112700), three MYB TFs (*OsCCA1*, Os08g0157600; *OsMyb1R*, Os02g0685200; and *OsLHY*, Os04g0583900), C2H2 zinc finger protein (*RZF71*, Os12g0583700), and NAC TF (*OsNAC110*, Os09g0552900), was significantly higher in *2a-2-1* and *8b-3-9* mutant lines than in WT plants. Conversely, three genes in cluster 2, encoding a G2-like TF PHYTOCLOCK 1 (*OsPCL1*, Os01g0971800), a heat stress TF (*OsHsfC1b*, Os01g0733200), and an RNA polymerase Rpb1 (*RPB1*, Os02g0152800), demonstrated the opposite trend. Furthermore, we observed marked decreases in the expression of five genes in cluster 3 in *1a-8-1* and *2a-2-1* plants, including genes encoding homeobox-leucine zipper protein HOX6 (*Oshox6*, Os09g0528200), a WRKY TF (*OsWRKY76*, Os09g0417600), Jumonji 717 (*JMJ717*, Os08g0508500), a msp one binder kinase activator-like 1A (*MOB1A*, Os03g0409400), and a chromatin remodeling factor (*CHR743*, Os08g0243866). They were positively correlated with expression of the *Pro13a.1* gene but negatively correlated with *Pro13a.3* and genes in the Pro13b-I/II subgroups. Conversely, two cytokinin signaling genes (*RR6*, Os04g0673300 and *RR10*, Os12g0139400) grouped into cluster 4 showed significantly increased expression in these lines (Figure 9A).

As above, qRT-PCR analysis was performed to validate the findings observed with RNA-Seq data (Figure 9B). We focused on eight candidate TF genes grouped into four clusters: *OsHsfB2b* and *OsNAC110* in cluster 1, *OsPCL1* and *OsHsfC1b* in cluster 2, Oshox6 and *OsWRKY76* in cluster 3, and *RR6* and *RR10* in cluster 4. In cluster 1, *OsHsfB2b* (Os08g0546800) and *OsNAC110* (Os09g0552900) exhibited significantly higher expression in lines *2a-2-1* and *8b-3-9*, while *OsPCL1* (Os01g0971800) and *OsHsfC1b* (Os01g0733200) in cluster 2 showed the opposite trend. In addition, *Oshox6* (Os09g0528200) and *OsWRKY76* (Os09g0417600) in cluster 3 displayed markedly lower expression in lines *1a-8-1* and *2a-2-1*, whereas *RR6* (Os04g0673300) and *RR10* (Os12g0139400), grouped in cluster 4, showed significantly higher expression in these lines. Pearson’s correlation analysis indicated a strong correlation between RNA-Seq values and qRT-PCR data for these TFs, demonstrating a high level of agreement between the two datasets (Appendix A).

## 3. Discussion

### 3.1. Morphological Differences between Starch Granules in WT and 13 kDa Prolamin-Knockout Rice Plants

In this study, 13 kDa prolamin content in rice seeds was down-regulated using CRISPR-Cas9 and sgRNA-pro13a to knockout expression of genes in the Pro13a-I subgroup or sgRNA-pro13b to knockout expression of genes in the Pro13b-I/II subgroups. Four mutant lines were generated and characterized, including *1a-8-1* (*pro13a.1/13a.2*-knockout), *2a-2-1* (*pro13a.1*-knockout), *4b-9-7* (*pro13b.1/13b.13*-knockout), and *8b-3-9* (*pro13b.3*-knockout). Total grain weight in the *13 kDa prolamin*-knockout mutant lines was significantly lower, likely because of the lower total starch content. Similar results were observed in GPGb-RNAi mutant lines in which genes encoding glutelin A, prolamin 13a, and globulin are simultaneously suppressed by the RNAi system [29]. Each gene encoding glutelin A1, B2, and C1 was edited by the CRISPR-Cas9 system, generating Glu-knockout lines that also displayed similar phenomena [38]. The seeds in the mutant lines showed distinct alterations in starch content, starch granule structure, and seed appearance, presenting with a “chalky” phenotype not observed in WT (Figure 4).

Starch synthesis is initiated by converting glucose-1-phosphate to ADP-glucose by ADP-glucose pyrophosphorylase (AGPase). ADP-glucose is used for amylose synthesis by granule-bound starch synthases (GBSSs) and for amylopectin synthesis via a series of enzymatic reactions such as soluble starch synthases (SSs), branching enzymes (BEs), and debranching enzymes (DBEs) [39]. In GPGb-RNAi and Glu-knockout lines, it has been presented that reduced starch content and altered starch composition are associated with a significant decrease in some starch synthesis genes at the transcriptional level [29,38]. In this study, the expression of *SSI* and *BEIIa* genes is significantly lower in the immature seeds of the four *13 kDa prolamin*-knockout mutant lines, which could explain the reduced total starch content. On the other hand, the up-regulation of the *GBSSI* gene in the immature seeds of the *4b-9-7* (*pro13b.1/13b.13*-knockout) line and its down-regulation in the immature seeds of other mutant lines are consistent with higher amylose content in starch granules in *4b-9-7* plants and lower amylose content in the other mutant plants. Moreover, rod/filamentous starch granules have been reported previously in grains with high amylose content [40,41], which is consistent with the idea that rod/filamentous starch granules observed in *4b-9-7* plants are caused by starch with increased amylose content.

### 3.2. Abnormal PB-I Morphology in 13 kDa Prolamin-Knockout Plants

In WT rice, PB-I particles form by sequential deposition of concentric circular layers of specific SSPs, where 10 kDa prolamin (Pro10) accumulates in the core layer, followed sequentially by cys-poor 13 kDa prolamin (Pro13b-I), cys-rich 13 kDa prolamin (Pro13a-I/II), 16 kDa prolamin (Pro16), and an outermost layer of cys-poor 13 kDa prolamin (Pro13b-II) [8,42]. It can be asserted that the specific interactions among rice prolamins play a crucial role in the formation of PB-I structures. Cys-rich prolamins (Pro10, Pro13a, and Pro16) can form intermolecular disulfide bonds in the layer surrounding the central core, while cys-poor 13 kDa prolamins (Pro13b-I/II) may interact with cys-rich prolamins through hydrophobic interactions [42,43]. It has been reported that the composition of prolamins in PB-I has a profound impact on its layered structure [44,45]. Previous studies reported aberrant PB-I morphology in rice seeds from 13 kDa-prolamin RNAi-targeted transgenic plants. Specifically, seeds with reduced expression of RM1 (former name of *Pro13a.2* in Pro13a-I subgroup), RM2 (former name of *Pro13b.2* in the Pro13b-I subgroup), RM4 (former name of *Pro13b.11/12/13* in the Pro13b-II subgroup), and RM9 (former name of *Pro13a.4* in the Pro13a-II subgroup) displayed PB-Is with cracks and jagged periphery structures [4]. In addition, small PB-Is with no lamella structure were observed when expression of the Pro13a-I subgroup was suppressed [2]. Furthermore, the ERAD protein degradation system plays a critical role during seed development, when the biosynthesis of SSPs is intensely up-regulated. ERAD polyubiquitinates and then helps eliminate misfolded, damaged, and unfolded proteins by a process involving the ubiquitin ligase complex composed of OsHrd1 and OsOS-9 [35,37,46,47]. The functional loss of OsHrd3 and the ubiquitin ligase complex is thought to play a role in forming deformed PB-Is [48], and reduced expression of OsDER1, a protein-interacting partner of OsHrd3, also led to the formation of cracked PB-Is. These results indicate that the quality of SSPs likely has a significant impact on PB-I structure. In this study, knockout of some of the genes in the Pro13a-I subgroup (*pro13a.1/13a.2* in *1a-8-1* line and *pro13a.1* in *2a-2-1*) and the Pro13b-I/II subgroups (*pro13b.1/13b.13* in *4b-9-7* and *pro13b.3* in *8b-3-9*) also led to notable structural changes in PB-I morphology. Compared to WT, mutant lines with reduced expression of Cys-rich Pro13a-I subgroup 13 kDa prolamins had small PB-I particles lacking lamellar spherical structure. In contrast, mutant lines with low expression of Cys-poor Pro13b-I/II subgroups had PB-I structures with a dark appearance, due to the changes in electron density (Figure 5A).

### 3.3. Functional Analysis of Genes Correlated with 13 kDa Prolamin Genes

In this study, subsets of 13 kDa prolamin genes were targeted/edited using CRISPR-Cas9, but compensatory changes in the expression of non-targeted genes were observed (Appendix A and Figure 5). Specifically, in the immature seeds of the *pro13a.1/13a.2*-knockout mutant line (*1a-8-1*), the transcription of genes in the Pro13a-II and Pro13b-I/II subgroups were up-regulated; in the *pro13a.1*-knockout mutant line (*2a-2-1*), the transcription of genes in the Pro13a-II subgroup was up-regulated but those in the Pro13b-I/II subgroups were not changed. Moreover, in the immature seeds in which transcription of the genes in the Pro13b-I/II subgroups were suppressed (*4b-9-7*), a weak up-regulation of genes in the Pro13a-I subgroup was observed but not changed in the Pro13a-II subgroup. In the *pro13b.3*-knockout mutant line (*8b-3-9*), down-regulation of the genes in the Pro13b-I subgroup was accompanied by up-regulation of genes in the Pro13a-II, Pro13b-II, and Pro10 groups. Moreover, to gain insight into molecular events that occur in mutants suppressing the expression of specific prolamin subgroups in more detail, correlation network analysis identified six clusters of DEGs, each of which displayed unique patterns of expression in the four mutant plant lines. The DEGs played roles in RNA processing, RNA transcription, transport, stress response, protein synthesis, protein targeting, protein posttranslational modification, protein folding, and protein degradation.

During rice seed development, large amounts of SSPs are translated and matured in ER, in which the accumulation of unfolded proteins activates unfolded protein response (UPR), leading to enhanced expression of genes encoding chaperones such as BiPs and PDILs to help proper protein folding, proteins involved in protein secretion to target organelles, and proteins implicated in ER-associated degradation (ERAD) of misfolded proteins [49]. Recent articles report that the suppression of SSP using RNAi and CRISPR-Cas9 systems can impact the expression of ER chaperones and co-chaperones such as BiPs, protein folding machinery (CNX, PDIs, and HSPs), and protein-homeostasis-related sHSPs, resulting in ER-stress and altered SSP composition [2,4,29,38]. Moreover, the overexpression of ER stress marker genes (*OsbZIP50*, *OsBiP1*, *OsBiP2*, and *OsBiP3*) suppresses starch synthesis-related genes such as *GPA3*, *FSE1*, *FLO7*, and *OsNF-YB1*, resulting in grain chalkiness and lower amylose and SSP contents [16]. The characterization of a rice opaque endosperm mutant, *opaque3* (*o3*), with over-accumulation of 57 kDa proglutelins and lower SSP and starch contents indicated that *O3* encodes OsbZIP60 that controls ER-homeostasis and grain quality by regulating genes involved in starch synthesis (*GBSSI*, *AGPL2*, *SBEI*, and *ISA2*), SSP (*OsGluA2*, *Prol14*, and *Glb1*), and chaperone (*OsBiP1* and *PDIL1-1*) [50].

In this study, we showed that nine ERAD-related genes encoding heat shock proteins (three *HSP70* and one *HSP90*), small heat shock proteins (*HSP26, HSP20, HSP17.7,* and *HSP22a*), and one co-chaperone (*OsDjC35*) were grouped in clusters 1 and 3 (Figure 8, Appendix A). Cluster 1 includes genes that are positively correlated with prolamin genes in the Pro13a-II and Pro13b-I/II subgroups, including three *OsHSP70* genes that are up-regulated in *pro13a.1*-knockout (*2a-2-1*) and *pro13b.3*-knockout (*8b-3-9*) mutant lines. Cluster 3 contains genes that are negatively correlated with prolamin genes in the Pro13a-II and Pro13b-I/II subgroups but positively correlated with the *Pro13a.1* gene in the Pro13a-I subgroup; these include *OsHSP90, OsHSP26, OsHSP20, OsHSP17.7, OsHSP22a,* and *OsDjC35*. The genes were only up-regulated in the *pro13b.1/13b.13*-knockout (*4b-9-7*) mutant line. HSP70 functions as a molecular chaperone, facilitating the proper folding of nascent polypeptides, refolding misfolded proteins, and contributing to substrate degradation via the ubiquitin-proteasome system [51,52]. HSP90, along with its co-chaperones, is involved in ERAD, a sophisticated cellular process responsible for identifying and degrading misfolded proteins through the ubiquitin-proteasome system [53,54,55]. The small heat shock proteins (sHSPs) transfer denatured proteins to the HSP 70/90 chaperone system to perform continuous ATP-dependent refolding of these proteins, preventing protein misfolding and aggregation [56]. DjC35, as a member of HSP40 co-chaperones (J-domain proteins), enhances HSP70 function by promoting ATPase activity, stabilizing substrate interactions, and preventing the aggregation of unfolded proteins [57,58,59]. Up-regulation of the genes involved in protein processing pathways in the ER probably indicates that *prolamin gene*-knockout affects ER chaperones and SSP composition. These results suggest that the expression of ERAD-related genes and SSPs varied in the four mutant plant lines, causing a variable impact on grain quality.

Recently, it has been reported that TFs spontaneously regulate starch and SSP synthesis during seed development in rice [12], including *OsbZIP58* [14,60], *Nhd1* [61], *OsNAC20* [15], *OsNAC26* [15], *FLO2* [62], and *OsMADS1* [63]. *OsbZIP58* activates the expression of starch synthesis-related genes (*OsAGPL3*, *GBSSI*, *SSIIa*, *SBE1*, *BEIIb*, and *ISA2*) by directly binding their promoters [60] and interacts with rice prolamin box binding factor, activating the expression of SSP genes (*GluA1*~*3*, *GluB1*, *GluD1*, and *10/13/16 kDa prolamins*) [14]. Up-regulation of N-mediated-heading-date (*Nhd1*), a MYB TF, resulted in reducing amylose and enhancing amylopectin through the down-regulation of *GBSSI* and up-regulation of *SSII-3*/*SSIIa* and *FLO5*, and suppressing the expression of *GluA2*, one of the SSP genes [61]. Wang’s group [15] reported that *OsNAC20* and *26* up-regulate the expression of SSP genes (*GluA1*, *GluB4/5*, *globulin,* and *16 kDa prolamin*) and starch synthesis-related genes (*SSI*, *Pul*, *AGPS2b*, *AGPL2,* and *SBEI*) together. The function-loss mutation of the *FLOURY ENDOSPERM2* (*FLO2*) gene caused a decrease in grain size and starch content by reducing the expression of starch and SSP-related genes, while its overexpression resulted in an increase in amylose content and grain weight/size by enhancing the expression of starch (*AGPL1*, *AGPL4,* and *GBSSI*) and SSP genes (*GluA1* and *globulin*) [62]. Moreover, Oat-like rice (Olr), which was generated by a mutated *OsMADS1* allele, showed a loose arrangement of starch granules, and reduced starch content but increased SSP content [63]. The results indicate that the regulatory mechanisms of SSP and starch synthesis are coupled by TFs, and investigating novel TFs and understanding their regulatory networks are crucial factors in improving seed quality and yield.

In this study, we provide the involvement of 17 TFs belonging to the correlation network affected by 13 kDa prolamin gene expression, most of which are responsive to abiotic stress. These TFs belong to clusters 1, 2, 3, and 4 in the correlation network of genes. There are seven TFs in cluster 1, including *OsHSFB2b, MADS26, CCA1, RZF71, OsMYB1R, OsNAC110,* and *LHY*. *OsHSFB2b* acts in response to heat stress, drought, and salt stress [64], while *MADS26* plays diverse roles in plant development, stress response, and pathogen resistance [65,66]. *CCA1* and *LHY* regulate *ABF3* expression and seed germination in response to salt stress [67], and *RZF71* also enhances tolerance to salinity and drought [68]. *NAC110* provides high tolerance to drought and salt stress via an ABA-independent pathway [69]. Conversely, expression of three TFs in cluster 2 (*RPB1, HSFC1b*, and *PCL1*) were negatively correlated with expression of *Pro13a.3* and genes in the Pro13b-I/II subgroups and were down-regulated in *pro13a.1*-knockout (*2a-2-1*) and *pro13b.3*-knockout (*8b-3-9*) mutant lines (Figure 9). *OsPCL1* positively regulates the response to cold stress [70], and *OsHsfC1b* is involved in ABA-mediated tolerance to salt stress [71]. The results indicate that TFs correlated with prolamin genes in Pro13a-II and Pro13b-I/II subgroups play a role in stress response, where knockout of *Pro13a.1* or *Pro13b.3* genes likely causes stress. Five TFs were identified in cluster 3, including *WRKY76, CHR743, JMJ717, MOB1A,* and *OsHox6*; expression of these TFs was down-regulated in *pro13a.1/13a.2*-knockout (*1a-8-1*), *pro13a.2*-knockout (*2a-2-1*), and *pro13b.3*-knockout (*8b-3-9*) mutant lines. *WRKY76*, a rice transcriptional repressor, has dual roles in blast disease resistance and cold stress tolerance [72]. *OsHox6,* belonging to the homeodomain leucine zipper (HD-Zip) protein sub-family I, is up-regulated under water-deficit conditions [73]. *JMJ717*, a member of the Jumonji C (jmjC) domain-containing proteins reversing histone methylation, is crucial for various biological processes, including plant defense [74]. *RR6,* a type-A cytokinin response regulator, is involved in hormone signaling and pathogen response, while *RR10*, a type-B response regulator, negatively regulates the response to salinity stress [75,76]. Cluster 4 includes two TFs that are positively correlated with the *Pro13a.3* gene and prolamin genes in the Pro13b-I/II subgroups but negatively correlated with the *Pro13a.1* gene. *RR6* and *RR10* were up-regulated in *pro13a.1/13a.2*-knockout (*1a-8-1*), and *pro13a.2*-knockout (*2a-2-1*) mutant lines. TFs in clusters 3 and 4 respond to biotic and abiotic stresses and hormones. These findings suggest that targeted editing of prolamin genes using CRISPR-Cas9 causes different types of stress and different stress responses in the mutant plants, depending on the target gene(s). These responses play a role in determining SSP content, the severity of compensatory effects, and the overall quality of the mutant seeds.

In conclusion, based on the high homology among genes in a prolamin subgroup, we attempted to knockout two and seventeen genes in prolamin 13a-I and 13b-I/II subgroups using each sgRNA, which was designed at a specific region conserved among genes of each prolamin subgroup. The editing of up to two of the seventeen genes in the prolamin 13b-I/II subgroup suggests the editing limitations of one sgRNA and the need to use multiple sgRNAs. The sgRNA design in conserved regions represents a very useful strategy for editing genes with redundant functions and high homology. Furthermore, we investigated the relationship between 13 kDa prolamin suppression via the CRISPR-Cas9 tool and morphological changes in starch granules and protein bodies using RNA-Seq analysis, suggesting the involvement of ER stress-responsive TFs and (co-)chaperones. Although understanding the regulatory mechanisms is very complex and exploring their upstream genes has been limited, this study will provide important information for molecular breeding to improve grain quality.

## 4. Materials and Methods

### 4.1. Design of sgRNA

In order to edit genes encoding prolamin in rice, two sgRNAs (sgRNA-pro13a and -pro13b) were designed to target conserved nucleotide sequences in Pro13a-I and Pro13b-I/II subgroup genes, respectively, using CRISPR-P 2.0, an online tool (http://crispr.hzau.edu.cn/CRISPR2/ (accessed on 30 March 2019)), as described in Appendix A. The sgRNA-pro13a targets *Pro13a.1* (Os07g0206400) and *Pro13a.2* (Os07g0206500). The sgRNA-pro13b targets 17 genes, including *Pro13b.1* (Os07g0219250), *Pro13b.2* (Os07g0219300), *Pro13b.3* (Os07g0219400), *Pro13b.4* (Os07g0220000), *Pro13b.5* (Os05g0328333), *Pro13b.7* (Os05g0328632), *Pro13b.8* (Os05g0328800), *Pro13b.9* (Os05g0328901), *Pro13b.10* (Os05g0329001), *Pro13b.11* (Os05g0329100), *Pro13b.12* (Os05g0329300), *Pro13b.13* (Os05g0329350), *Pro13b.14* (Os05g0329400), *Pro13b.15* (Os05g0329700), *Pro13b.16* (Os05g0329200), *Pro13b.17* (Os05g0330150), and *Pro13b.18* (Os05g0330600) (Figure 1A and Appendix A). The sequence alignment of all 13 kDa prolamin genes is shown in Appendix A.

The efficiency of the designed sgRNAs was tested in vitro using the Guide-it sgRNA In Vitro Transcription and Screening System. The designed sgRNAs were produced by in vitro transcription reactions according to the instructions of the Guide-it sgRNA In Vitro Transcription Kit (Cat. #632635, Takara Bio, Shiga, Japan). DNA templates were synthesized by PCR with specific primers (Appendix A). Then, the synthesized DNA template, sgRNA, and recombinant Cas9 nuclease were combined as indicated in the Guide-it sgRNA screening kit (Cat. #632639, Takara) and incubated at 37 °C for 1 h. Reaction products were analyzed by agarose gel electrophoresis (Appendix A).

### 4.2. Cloning CRISPR-Cas9 Vector and Agrobacterium-Mediated Generation of 13 kDa Prolamin-Knockout Rice Plants

To insert the designed sgRNAs into the pRGE31 vector, two single-stranded 70 nt DNA oligonucleotides (ssDNA oligo) containing 20 nt sgRNA sequences (Appendix A) were designed, synthesized, and purified with a final concentration of 0.2µM. The pRGE31 backbone vector was digested with *Bsa* I (Cat. #R3773, NEB, Ipswich, MA, USA) for 16 h at 37 °C. A reaction mixture consisting of 5 µL of ssDNA oligo (0.2 µM), 30 ng of restriction enzyme-linearized vector, and 10 µL of NEBuilder HiFi DNA Assembly Master Mix (Cat. #E2621, NEB) with a final volume of 20 µL was incubated for 1 h at 50 °C for ligation and then transformed into *E. coli* DH5-α cells. The recombinant clones were extracted and sequenced with the undigested pRGE31 as the negative control and the insertion donor as the positive control, as previously described [38]. Furthermore, to construct binary vectors for *Agrobacterium*-mediated rice transformation, the sgRNA cassettes and pCAMBIA-Cas9 binary vector were double-digested with *Hind* III (Cat. #R3104, NEB,) and *Bgl* II (Cat. #R0144S, NEB) for 5 h at 37 °C, separated by gel electrophoresis, extracted, and then ligated with the ratio of vector and insert of 1:10 using T4 DNA ligase (Cat. #B0202A, NEB) at 16 °C for 13 h (Appendix A). Finally, the ligated binary vector was confirmed by colony PCR and DNA sequencing, using the undigested pCAMBIA-Cas9 as the negative control and the insertion cassettes as the positive control, with primers listed in Appendix A.

Finally, the constructed binary vectors (Figure 1B) were transformed into the *Agrobacterium tumefaciens* strain (EHA105) and co-cultured with the calli induced from mature seeds of rice (*Oryza sativa* L. *Japonica* cv. ‘Ilmi’). Then, the selection and regeneration of transformed calli were carried out as previously described [77].

### 4.3. Screening Transgenic Rice Plants

In order to detect mutations in transgenic rice plants, genomic DNA were extracted from the leaves (50 mg) of thirty-day-old rice plants as previously described [78]. *SpCas9* and *HygR* genes were amplified by PCR to screen for the presence/absence of the binary vector transformed using gene-specific primers (Appendix A), and reaction products were analyzed by agarose gel electrophoresis (Appendix A). Furthermore, the targeted genes were amplified by PCR with specific primers for deep-sequencing (listed in Appendix A), sequenced by the Mini-seq service (Bio-Core center, KAIST, Korea), and then analyzed using RGEN tools (http://www.rgenome.net/ (accessed on 9 January 2024)). Mutation frequency was calculated based on the percentage of sequences with mutations (insertion or deletion of nucleotides) over the total number of sequences sequenced. We calculated that mutation frequency from deep sequencing data and compared the target regions in a sample of T_0_, T_1_, and T_2_ plants.

### 4.4. RNA Isolation and qRT-PCR

Total RNA was isolated from three immature rice seeds at 2 WAF according to a previously published protocol [79]. The cDNA was synthesized using 1 µg RNA and the QuantiTect Reverse Transcription Kit (Cat.#205311, Qiagen, Hilden, Germany) according to the manufacturer’s instructions. qRT-PCR was performed as described in the QuantiTect SYBR Green PCR Kit (Cat.#204343, Qiagen, Hilden, Germany) in 20 µL reactions containing 5 ng cDNA, 0.5 µM forward primer, 0.5 µM reverse primer, and 2X Master Mix (10 µL). The Qiagen Rotor-gene Q cycler was used with the following settings: an initial denaturation at 95 °C for 10 min, 40 PCR cycles at 95 °C for 30 s, 60 °C for 30 s, and 72 °C for 30 s, a final extension at 72 °C for 10 min, and a melting curve analysis from 72 to 95 °C. The primers for qRT-PCR are listed in Appendix A. Expression levels were normalized using the 2^−ΔΔCT^ method with *Ubiquitin* (Os02g0161900) as the reference gene and WT as the reference sample [80] and represented as the log_2_ value of the average fold change calculated from three biological replicates.

### 4.5. Total Seed Storage Protein Extraction, SDS-PAGE, and Western Blot Analysis

To extract total SSP in rice, three de-husked mature seeds were ground in a mortar and pestle using liquid nitrogen and mixed with 1 mL of SDS-Urea buffer (250 mM Tris-HCl, pH 6.8, 4% SDS, 8 M urea, 20% glycerol, and 5% β-mercaptoethanol (2-ME) [37]. Then, total SSP protein (5 µg) was separated by gradient SDS-polyacrylamide gel electrophoresis (10–17.5%). Proteins were stained with Coomassie brilliant blue (CBB). For western blot analysis, proteins were transferred from the gel to a polyvinylidene difluoride (PVDF) membrane (WestPure PVDF Membrane 0.45 μm, Cat. #LC7032-300, GenDEPOT, Katy, TX, USA) using a semi-dry blotter (Sigma-Aldrich, Saint Louis, MO, USA) as previously described [81]. The membrane was incubated in the diluted (1:2000) Pro13a.2 (Cat. #PHY4308A, PhytoAB, San Jose, CA, USA) and Pro13b.1/2 (Cat. #PHY4440S, PhytoAB), antibodies in PBS buffer (10 mM Na_2_HPO_4_, 1.8 mM KH_2_PO_4_, 137 mM NaCl, 2.7 mM KCl, pH 7.4) containing 2% bovine serum albumin (BSA) for 3 h, washed three times in PBS buffer for 5 min, incubated in diluted (1:10,000) anti-rabbit IgG, alkaline phosphatase-linked secondary antibody (Cat. #S3731, Promega, Madison, WI, USA) in PBS buffer containing 2% BSA for 1 h, and then washed three times in PBS buffer for 5 min. Proteins were visualized using the NBT/BCIP buffer (Cat. #34070, Thermo Fisher scientific, Waltham, MA, USA) system according to the manufacturer’s instructions.

### 4.6. SSP Fractionation

Albumin/globulin, prolamin, and glutelin proteins were fractionated based on their solubility using different solvents as previously described [38,82]. Protein content in each fraction was determined by bicinchoninic acid (BCA) assay (Pierce BCA Protein Assay Kit, Cat. #BCA1, Sigma-Aldrich) according to the manufacturer’s instructions.

### 4.7. Measurement of Rice Agronomic Traits and Starch Content

Rice agronomic traits were quantified, including the average weight of 100 grains and the average length, width, and thickness of one grain. Ten de-husked mature seeds were ground in a mortar and pestle using liquid nitrogen. Starch and amylose content in 25 mg grain powder were measured using an Amylose/Amylopectin Assay Kit (Cat.#K-AMYL, Megazyme, Wicklow, Ireland) according to the manufacturer’s instructions.

### 4.8. Microscopic Analysis

To observe the morphology of starch granules in rice endosperm, dry, mature seeds were transversely cut with a razor blade, mounted on SEM stubs, and coated with platinum particles. The mounted specimens were examined by a ZEISS Gemini 500 Scanning Electron Microscope (SEM) (Carl Zeiss AG, Jena, Germany) at 15 kV, as previous described [38]. To observe PB-I and PB-II particles, immature rice seeds at 2 WAF were fixed in 50 mM cacodylate buffer (pH 7.2) containing 2% glutaraldehyde and 2% paraformaldehyde for 4 h at room temperature, washed three times in 50 mM cacodylate buffer (pH 7.2) for 30 min, fixed in 50 mM cacodylate buffer (pH 7.2) containing 1% OsO4 for 1 h, washed three times in 50 mM cacodylate buffer (pH 7.2) for 30 min, dehydrated in a gradient ethanol series (30, 50, 70, 90, 95, and 100%) at 30 min intervals, and then sequentially immersed in a mixture of LR White resin and 100% ethyl alcohol in ratios of 1:2, 1:1, 2:1, and 1:0 at 60 °C and 6 h intervals. Samples with LR White resin in a gelatin capsule were cured at 60 °C for 24 h, cut to 80–100 µm thickness using an ultra-microtome, mounted on carbon-coated nickel grids, double-stained with 4% uranyl acetate and 0.4% lead citrate, and observed with a JEM-2100F transmission electron microscope (TEM) (JEOL Ltd., Tokyo, Japan).

### 4.9. RNA Sequencing

RNA sequencing analysis was performed by DNALINK biotechnology company (Seoul, Republic of Korea) using total RNA extracted from immature rice seeds 2 WAF. The quality of total RNA was analyzed using an Agilent 2100 Bioanalyzer Expert (Agilent Technologies, Inc., Waldbronn, Germany). Data were analyzed using a Kallisto tool to align RNA-seq reads with the rice reference genome (*Nipponbare* rice genome IRGSP-1.0) [83]. RSEM software (v1.3.3) was used to quantify each transcript [84], and the DESeq2 method from the edgeR software (https://bioconductor.org/packages/release/bioc/html/edgeR.html (accessed on 10 February 2024)) used to identify DEGs [85,86].

### 4.10. Transcriptome Analysis and Network Visualization

DEGs were annotated according to the databases of the Gramene Mart of Gramene (https://www.gramene.org/ (accessed on 10 February 2024)) linked with *Oryza sativa* Japonica group genes (IRGSP-1.0) and Osa_RAPDB (Rice Annotation Project Database) reference map of *Oryza sativa* Japonica group (RAPDB-IRGSP1.0) in MapMan (https://mapman.gabipd.org/home (accessed on 10 February 2024)). Venn diagrams of up- and down-regulated DEGs were designed using Venny2.1 (https://csbg.cnb.csic.es/BioinfoGP/venny.html (accessed on 10 February 2024)). A Pearson correlation test was performed, and highly correlated genes (|r| > 0.7) were selected and visualized using Cytoscape software (ver. 3.10.1). Genes involved in RNA processing, RNA transcription, protein synthesis, stress, and transport were correlated with 13 kDa prolamin genes. Expression of candidate genes involved in protein processing in the ER and TFs were visualized as a heatmap using PermutMatrix software (ver. 1.9.3).

### 4.11. Statistical Analysis

A student’s *t*-test (two samples assuming equal variances) was conducted using Excel (Microsoft office 365), which is used to compare means between two independent groups. For each comparison, *p*-values were calculated to determine statistical significance. Significant differences are indicated with asterisks (*), where * *p* < 0.1, ** *p* < 0.01, and *** *p* < 0.001. This analysis ensures that the reported differences in seed characteristics, starch composition, protein composition, and expression genes are statistically validated.

## Figures and Tables

**Figure 1 ijms-25-06579-f001:**
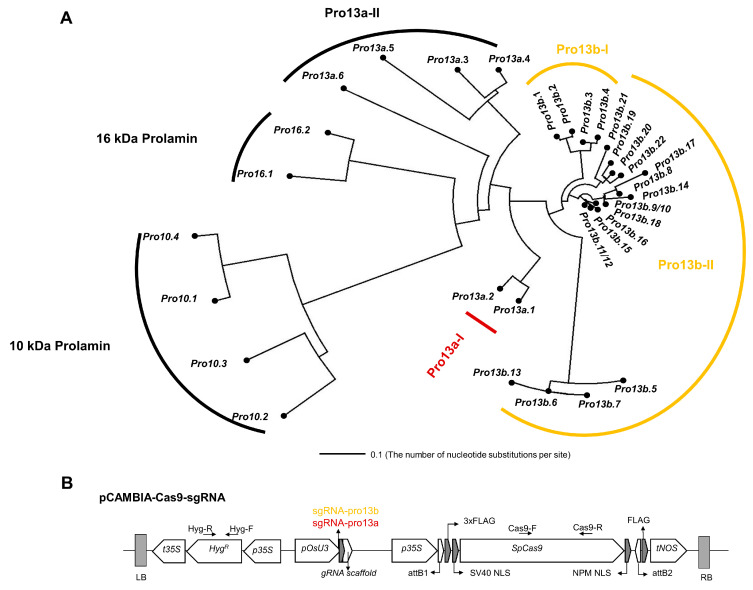
Strategy for simultaneous editing of multiple prolamin genes using CRISPR-Cas9. (**A**) A phylogenic tree of prolamin genes was constructed using Dendroscope (ver 3.5.7). Genomic sequences were from the Rice Annotation Project Database (https://rapdb.dna.affrc.go.jp/, (accessed on 30 March 2019)). Target genes of two sgRNAs, sgRNA-pro13a and sgRNA-pro13b, are shown in red and yellow, respectively. (**B**) Structure of the pCAMBIA-Cas9-sgRNA binary vector. The binary vector carries sgRNA-pro13a (AACGTAGCTGGTTGCCAGAAGGG) or sgRNA-pro13b (AAACGCAGCTGATTGCAAGAAGG) under the control of the pOsU3 promoter. Primers for PCR amplification are indicated by black arrows.

**Figure 2 ijms-25-06579-f002:**
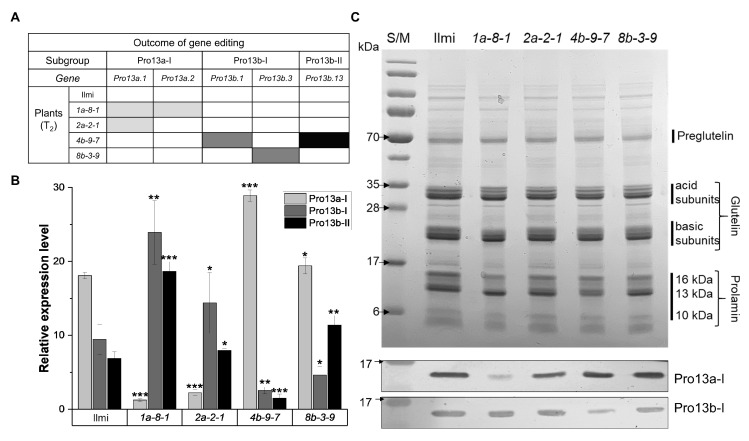
Production of *13 kDa prolamin*-knockout T_2_ lines with reduced prolamin content. (**A**) Outcomes of gene editing in the T_2_ generation of *13 kDa prolamin*-knockout lines (*1a-8-1*, *2a-2-1*, *4b-9-7*, and *8b-3-9*). Shaded boxes indicate which genes in each mutant line carry frameshift mutations. Different colors represent mutations in different 13 kDa prolamin subgroups. (**B**) Comparison of Pro13a-I, Pro13b-I, and Pro13b-II subgroup expression in WT (Ilmi) and *13 kDa prolamin*-knockout lines. qRT-PCR was performed 2 weeks after flowering to quantify transcripts of the indicated 13 kDa prolamin genes in immature seeds of WT and mutant plants. Values are mean ± SD (n = 3). *p*-values were calculated by the Student’s *t*-test (* *p* < 0.1, ** *p* < 0.01, and *** *p* < 0.001). (**C**) Analysis of SSP levels in Ilmi and *13 kDa prolamin*-knockout lines by SDS-PAGE (top) and Western blot (below). Total SSPs (5 µg) were separated on a gradient SDS-PAGE gel (10–17.5%). The resolved SSPs were transferred onto a polyvinylidene fluoride (PVDF) membrane, and the membrane was incubated with anti-Pro13a-I or anti-Pro13b-I antibodies, as indicated. S/M: Size marker.

**Figure 3 ijms-25-06579-f003:**
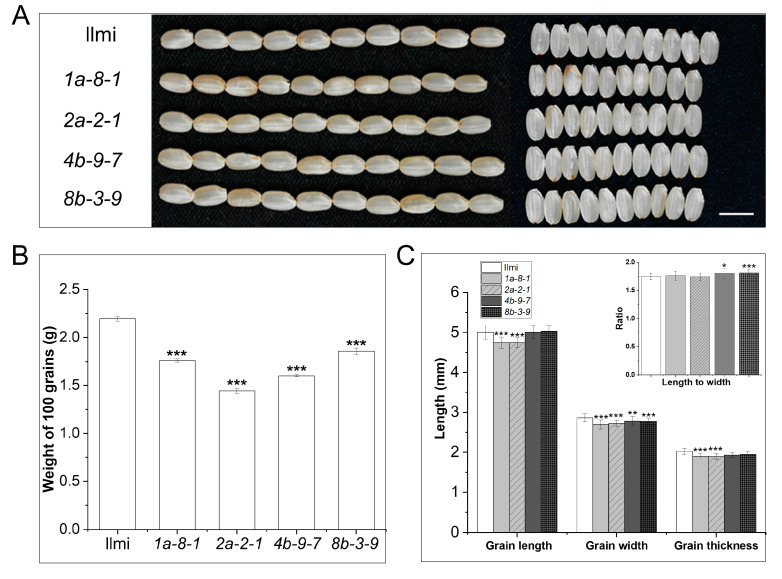
Morphology of seeds from *13 kDa prolamin*-knockout plants. (**A**) Representative images of WT and knockout seeds. Left: length; Right: width of de-husked seeds. Scale bar = 5 mm. (**B**) Weight of 100 grains of Ilmi and *13 kDa prolamin*-knockout seeds. Values are mean ± SD (n = 3). (**C**) Comparison of length, width, thickness, and length-to-width ratio of grain between Ilmi and *13 kDa prolamin*-knockout seeds. Values are mean ± SD (n = 30). *p*-values were calculated by the Student’s *t*-test (* *p* < 0.1, ** *p* < 0.01, and *** *p* < 0.001).

**Figure 4 ijms-25-06579-f004:**
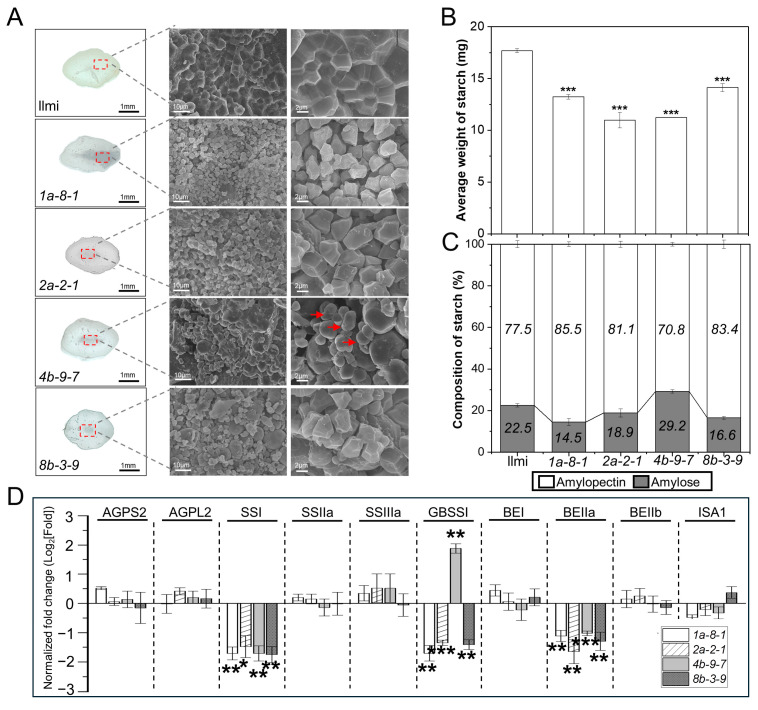
Starch granule appearance and starch content in WT and *13 kDa prolamin*-knockout seeds. (**A**) Representative images of seed transverse sections showing the structure of starch granules. The red arrow indicates rod/filamentous granules. (**B**) Average weight of starch per grain. (**C**) Relative composition of amylose and amylopectin. (**D**) Transcripts of genes involved in starch metabolism were analyzed by qRT-PCR. Normalized expression of target genes was calculated using the 2^−ΔΔCT^ method and is shown as a log_2_ value. Error bars represent ± SD of three replicates. *P*-values were calculated using the Student’s *t*-test (* *p* < 0.1, ** *p* < 0.01, and *** *p* < 0.001).

**Figure 5 ijms-25-06579-f005:**
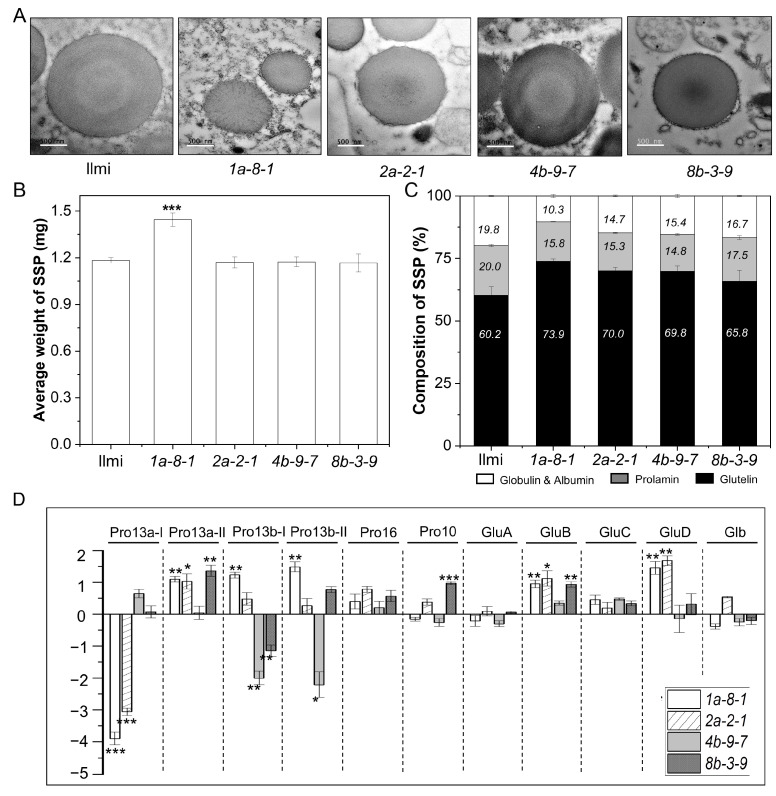
Structure of PB-I and composition of SSPs in Ilmi WT and *13 kDa prolamin*-knockout lines. (**A**) TEM image of PB-I in developing endosperm at 2 WAF. (**B**) Average weight of SSP per grain. (**C**) Relative composition of SSPs (Albumin/globulin, Prolamin, and Glutelin). (**D**) qRT-PCR analysis of SSP genes in *13 kDa prolamin*-knockout lines. Transcripts encoding different SSPs were analyzed in the immature seeds of WT and mutant lines. Normalized expression of the target genes was calculated using the 2^−ΔΔCT^ method and is represented as a log_2_ value. Error bars denote ± SD of three replicates. *p*-values were calculated using the Student’s *t*-test (* *p* < 0.1, ** *p* < 0.01, and *** *p* < 0.001).

**Figure 6 ijms-25-06579-f006:**
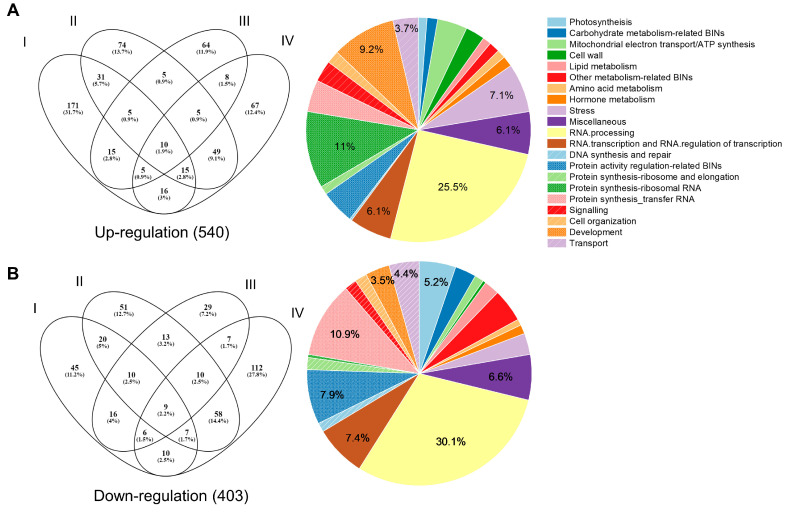
Transcriptome analysis identified DEGs in *13 kDa prolamin*-knockout lines. (**A**) Venn diagram and functional classification of 540 genes up-regulated relative to WT control (Ilmi). (**B**). Venn diagram and functional classification of 403 genes down-regulated relative to WT control (Ilmi). Results for *1a-8-1*, *2a-2-1*, *4b-9-7,* and *8b-3-9* plants are labeled with ellipses designated I, II, III, and IV, respectively.

**Figure 7 ijms-25-06579-f007:**
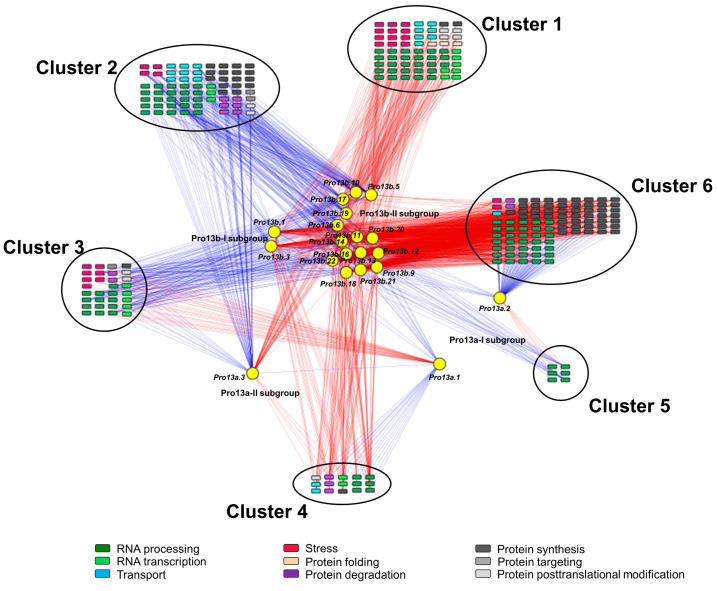
Correlation analysis among DEGs and 13 kDa prolamin genes. A Pearson correlation test was performed for DEGs and 13 kDa prolamin genes. The inclusion criteria was a correlation coefficient (r) ≥|0.7|. Transcripts were identified in nine functional groups: RNA processing, RNA transcription, transport, stress, protein synthesis, protein targeting, protein posttranslational modification, protein degradation, and protein folding. The data were visualized using Cytoscape software (ver. 3.10.1). Circles indicate 13 kDa prolamin genes, and squares indicate correlated genes. The blue line indicates a negative correlation, and the red line indicates a positive correlation.

**Figure 8 ijms-25-06579-f008:**
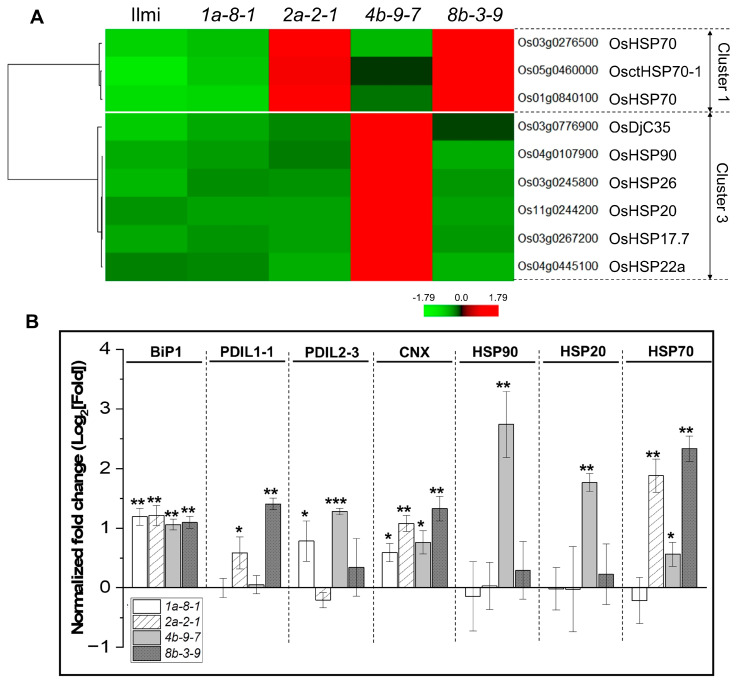
Expression of ER pathway genes in *13 kDa prolamin*-knockout lines. (**A**) Relative expression of DEGs involved in ER-related pathways in *13 kDa prolamin*-knockout lines. Normalized expression of each gene is represented by the Z-score. (**B**) qRT-PCR analysis of chaperone genes (BIP, PDIs, CNX, HSP90, HSP20, and HSP70) in *13 kDa prolamin*-knockout lines. Normalized expression of the target genes was quantified using the 2^−ΔΔCT^ method, and the relative expression of each gene is represented as a log_2_ value. Error bars denote ± SD of three replicates. *p*-values were calculated using the Student’s *t*-test (* *p* < 0.1, ** *p* < 0.01, and *** *p* < 0.001).

**Figure 9 ijms-25-06579-f009:**
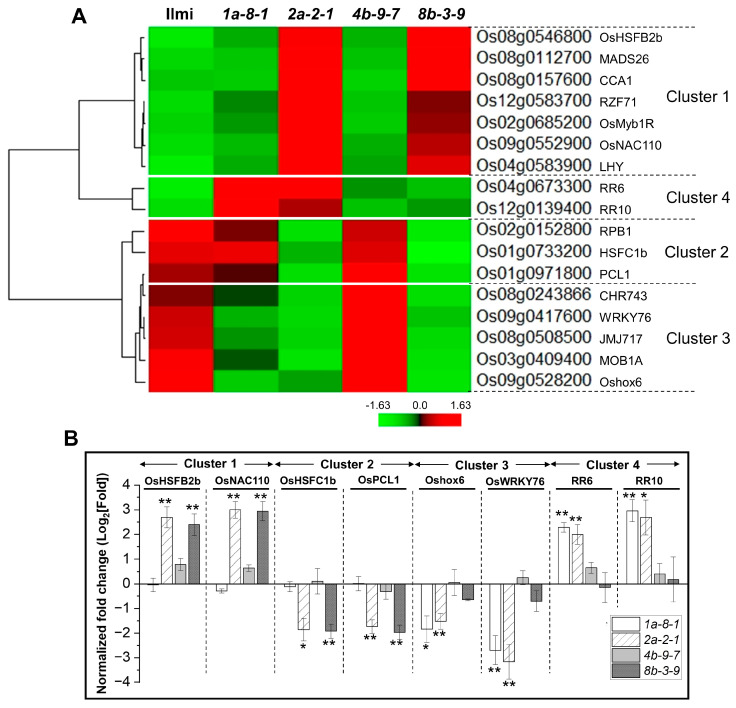
Expression of TF genes associated with 13 kDa prolamins. (**A**) The heatmap displays the expression profiles of the TF genes. This map shows altered expression of genes in *13 kDa prolamin*-knockout lines relative to WT. Normalized expression of each gene is represented by the Z-score. (**B**) Confirmation of the expression patterns of a few putative TF genes by qRT-PCR. Transcripts of TF genes were analyzed in the immature seeds. Normalized expression of the target genes was calculated using the 2^−ΔΔCT^ method, and is represented as a log_2_ value. Error bars denote ± SD of three replicates. *p*-values were calculated by the Student’s *t*-test (* *p* < 0.1 and ** *p* < 0.01).

## Data Availability

Data are contained within the article and Appendix A.

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
