# Peer review of "Compensatory Modulation of Seed Storage Protein Synthesis and Alteration of Starch Accumulation by Selective Editing of 13 kDa Prolamin Genes by CRISPR-Cas9 in Rice"

_ijms, 2024, doi:10.3390/ijms25126579_

Round 1

Reviewer 1 Report

Comments and Suggestions for Authors

In this study, the authors generated knockout rice of all the 13 kDa prolamins genes and performed a large of works to investigate its effects. The experiments are well designed and the results are interesting. I only have some minor concerns about this work.

1)      The title needs to be revised. In my view, knockout of the 13 kDa prolamins leads to only the compensatory modulation of seed storage protein synthesis but not the starch accumulation.

2)      In the Abstract, the authors mentioned that “These four mutant rice lines also demonstrated compensatory expression of glutelins and non-targeted prolamins, which resulted in low grain weight, altered starch content and atypically-shaped starch granules and protein bodies”. There was no evidence showed that low grain weight, altered starch content and atypically-shaped starch granules and protein bodies was due to the compensatory expression of glutelins and non-targeted prolamins.

3)      The authors should provide a supplementary file to show the sequence alignment of all the 13 kDa prolamins genes.

4)      Section 2.2, the authors used the CRISPR technique but not the T-DNA.

5)      In section 2.2, it is better to show the sequencing results of different mutant position in the genes in the supplementary files.

6)      Expression of genes in WT plants should be shown in Figs 4D and 5D.

7)      For the transcriptomic analysis, it is essential to look at the genes with the same expression among the four mutant rice.

Author Response

All your comments have been thoroughly addressed and the manuscript was revised accordingly. Please refer to the attachment Response to Reviewer 1

Reviewer 2 Report

Comments and Suggestions for Authors

I have reviewed the article entitled ‘Compensatory Modulation of Seed Storage Protein Synthesis and Starch Accumulation by Selective Editing of 13 kDa Prolamin Genes in Rice’ submitted by Pham and colleagues. The topic is interesting, and the article is well written. However, the following points should be discussed for better clarity.

-- The title can be revised by including keywords used in the methodology section, such as "CRISPR-Cas9" or "transcriptome analysis," to enhance the relevance in academic databases. The current title does not provide specific information about the methods or results.

-- In the abstract section, the authors provide a brief introduction of the objectives and findings of the study but lack context regarding the significance of the research within the broader field of plant biology or crop improvement. In this section, I recommend adding quantitative data to strengthen the credibility of the results. Also, please add a brief conclusion that could help readers understand the broader implications of this study.

-- In the introduction section, please add a comprehensive review of existing literature on the topic. No doubt, authors have discussed the use of CRISPR-Cas9 technology, however, doesn't provide much information on the experimental design or rationale behind targeting specific prolamin genes. Moreover, please add some lines on the specific research objectives or hypotheses.

-- In the M&M section, please add some information on critical parameters such as temperature, incubation times, and concentrations, which are essential for reproducibility. Please provide more comprehensive validation protocols and criteria to ensure the reliability of the experimental results. What are the positive and negative controls in the experiment? Please provide this information to validate the accuracy and specificity of the CRISPR-Cas9 editing and transformation processes.

--- Please add detailed statistical analysis, especially regarding the significance of the observed differences. Moreover, the authors have discussed the changes in seed morphology and starch composition, but the broader physiological consequences of these changes are not fully explored. Please elaborate.

---Revise the discussion section please discuss how the current findings align with or diverge from previous research; it would enhance the paper's contribution to the field. Also, discussing the potential role of specific prolamin subgroups or interactions between prolamins and other seed storage proteins in influencing granule morphology would provide a more comprehensive understanding. Please elaborate on how do changes in gene expression levels translate into modifications in seed composition, quality, and stress response mechanisms. Please add the potential limitations of the study, such as the specificities and off-target effects of CRISPR-Cas9 editing, as well as the need for additional validation of the observed phenotypic and transcriptional changes.

Specific comments

---Figure 2B, are the effects were non-significant?

---Lines 53-56: Give reference

---Line 69: ‘research’ change to ‘studies’

---Lines 90-91: fix typo error

---Line 129: why highlighted?

---References should be according to the journal style 

Comments on the Quality of English Language

NA

Author Response

All your comments have been thoroughly addressed and the manuscript was revised accordingly. Please refer to the attached Response to Reviewer 2 for details.

Reviewer 3 Report

Comments and Suggestions for Authors

Dear authors,

It's my pleasure to review this manuscript investigating engineering rice seed storage components using genome editing. Please address the following comments for the consideration of acceptance.

1. Since you mentioned the high sequence similarity of the prolamin genes, why not design sgRNA to target more genes or apply multiplexed sgRNAs to target more genes to achieve stronger mutants (more genes knockout in the family)?

2. As for the data in Table S2 and Figure S4A, please explain how you calculate the mutation frequency. Did you analyse the mutation frequency at each target site in T0 plants? 

3. Could you please explain why the increased average weight per grain was only observed in 1a-8-1 based on the RNA-seq data?

Author Response

All your comments have been thoroughly addressed and the manuscript was revised accordingly. Please refer to the attached Response to Reviewer 3 for details.

Round 2

Reviewer 2 Report

Comments and Suggestions for Authors

All the comments by all the reviewers have been incorporated at the appropriate place. Few more recent publications from IJMS on effects of these in recent years can add a great weightage. 

Comments on the Quality of English Language

Some sentences can be improved.